# SV40 miR-S1 and Cellular miR-1266 Sequester Each Other from Their Targets, Enhancing Telomerase Activity and Viral Replication

**DOI:** 10.3390/ncrna8040057

**Published:** 2022-07-28

**Authors:** Tetsuyuki Takahashi, Hirona Ichikawa, Yukiko Okayama, Manami Seki, Takao Hijikata

**Affiliations:** Department of Anatomy and Cell Biology, Research Institute of Pharmaceutical Science, Faculty of Pharmacy, Musashino University, Tokyo 202-8585, Japan; te_taka@musashino-u.ac.jp (T.T.); hiroichi@musashino-u.ac.jp (H.I.); tet19770729@gmail.com (Y.O.); s1743073@stu.musashino-u.ac.jp (M.S.)

**Keywords:** miRNA, simian virus 40, miR-S1, miR-1266, interplay

## Abstract

Virus-encoded microRNAs (miRNAs) target viral and host mRNAs to repress protein production from viral and host genes, and regulate viral persistence, cell transformation, and evasion of the immune system. The present study demonstrated that simian virus 40 (SV40)-encoded miRNA miR-S1 targets a cellular miRNA miR-1266 to derepress their respective target proteins, namely, T antigens (Tags) and telomerase reverse transcriptase (TERT). An in silico search for cellular miRNAs to interact with viral miR-S1 yielded nine potential miRNAs, five of which, including miR-1266, were found to interact with miR-S1 in dual-luciferase tests employing reporter plasmids containing the miRNA sequences with miR-S1. Intracellular bindings of miR-1266 to miR-S1 were also verified by the pull-down assay. These miRNAs were recruited into the Ago2-associated RNA-induced silencing complex. Intracellular coexpression of miR-S1 with miR-1266 abrogated the downregulation of TERT and decrease in telomerase activity induced by miR-1266. These effects of miR-S1 were also observed in miR-1266-expressing A549 cells infected with SV40. Moreover, the infected cells contained more Tag, replicated more viral DNA, and released more viral particles than control A549 cells infected with SV40, indicating that miR-S1-induced Tag downregulation was antagonized by miR-1266. Collectively, the present results revealed an interplay of viral and cellular miRNAs to sequester each other from their respective targets. This is a novel mechanism for viruses to manipulate the expression of viral and cellular proteins, contributing to not only viral lytic and latent replication but also cell transformation observed in viral infectious diseases including oncogenesis.

## 1. Introduction

Simian virus 40 (SV40) is a member of the Polyomaviridae family of tumor viruses. It has a circle of double-stranded DNA encoding viral regulatory proteins of large and small T antigens, agnoproteins, capsid proteins VP1–3, and an miRNA named miR-S1. Large and small T antigens (LTag and STag) have been extensively studied to provide fundamental insights into viral replication, cell transformation, and oncogenesis [1,2]. The large T antigen promotes not only viral DNA replication but also cell cycle progression and antiapoptosis of host cells by its action on the Rb proteins and by binding to p53 and preventing transcription of p53-dependent genes [3]. By binding to protein phosphatase 2 (PP2A) and reducing its function, the small T antigen leads to cell transformation and oncogenesis [4,5,6,7]. Interestingly, these large and small T antigens are the only identified targets of SV40′s miR-S1, which downregulates them [8,9]. Their downregulation by miR-S1 raises the question of whether the functional role of miR-S1 is to restrain viral replication and cell transformation. However, the function or roles of miR-S1 in viral replication and cell transformation remain largely unexplored.

Viral noncoding RNAs interact with host miRNAs to influence a variety of biological events, such as viral replication, host immune evasion, and cellular transformation [10,11]. The hepatitis C virus (HCV) 5′ UTR interacts with the liver-specific miR-122, stimulating viral replication [12]. V-ncRNAs derived from herpesvirus saimiri (HVS) have been demonstrated to interact with and downregulate host miRNAs, including miR-27, miR-16, and miR-142-3p, resulting in the upregulation of the miRNA targets [13]. The Epstein–Barr virus (EBV) circRNAs sponge human miRNAs, such as miR-31, miR-205, and miR-451, promoting cell proliferation and antiapoptosis [14]. However, nothing is known regarding whether viral miRNAs also act as sponges for host miRNAs to contribute to cellular transformation and viral replication. An miRNA–miRNA interaction has been experimentally shown in two mature cellular miRNAs, namely, miR-107 and let-7 [15], but not in viral or host miRNAs.

The present study evaluated the interaction of SV40 miR-S1 with cellular miRNAs and then explored whether and how the interaction between viral and cellular miRNAs promotes cell transformation and viral replication. TERT and small T antigen are upregulated when viral miR-S1 and cellular miR-1266-5p interact and sequester one another from their targets. The increased expression of TERT and small T antigen co-operatively enhances TERT activity. Viral replication and release are also increased in miR-1266-expressing cells compared with control cells, although control and miR-1266-expressing cells represent similar expression levels of large T antigens. The present study illustrates a novel interplay between viral and cellular miRNAs underlying the functional roles of miR-S1 in cell transformation and viral replication.

## 2. Materials and Methods

### 2.1. Cell Cultures

Cell lines used in this study and their culture conditions are described in the Appendix A

### 2.2. Mammalian Expression Vectors

In this report, miR-S1 expression vectors (pSRQ-CMV-miR-S1-PGK-puro and pQCXIH-miR-S1), an miR-1266 expression vector (pSRQ-CMV-miR-1266-PGK-puro), a TERT-HA expression vector (pZac-hTERT-HA-3′ UTR), and luciferase reporter vectors (pRL-Luc-miR-S1-3p compl., pRL-Luc-miR-S1-3p compl. mt, pRL-Luc-miR-138-5p, pRL-Luc-miR-152-3p, pRL-Luc-miR-337-5p, pRL-Luc-miR-434-5p, pRL-Luc-miR-532-3p, pRL-Luc-miR-921, pRL-Luc-miR-1266-5p, pRL-Luc-miR-4261, pRL-Luc-miR-6771-5p, pRL-Luc-hTERT-3′ UTR, and pRL-Luc-hTERT-3′ UTR mt) were constructed. The construction procedure is described in the Appendix A.

### 2.3. Retroviral Infection

To obtain stable expressing lines, BxPC-3 (miR-S1) and A549 (miR-1266, miR-S1, and both expressing) cells were subjected to retroviral infection. Using polyethylenimine MAX (Warrington, PA, USA), GP2-293 cells were cotransfected with 2 µg of miRNA expression vectors and 2 µg of pAmpho (Takara Bio USA, Mountain View, CA, USA). After 48 h, culture media containing retroviral particles were collected, mixed with 8 µg/mL polybrene, and then administrated to target cells. Infected cells were then selected with 1 µg/mL puromycin (Wako, Osaka, Japan) and 500 µg/mL hygromycin B (Wako, Osaka, Japan) for two days. The antibiotic-resistant cells were then used in further experiments.

### 2.4. Luciferase Reporter Assay

Using TransIT-293 (Mirus, Madison, WI, USA) for HEK293 and Lipofectamine 3000 (Invitrogen, Carlsbad, CA, USA) for miR-1266-expressing A549, cells were cotransfected with the internal pGL control vector, pRL-based reporter vectors, or miRNA expression vectors at a molar ratio of 1:5 or 1:5:25 (total 300 or 1550 ng vectors), respectively. After 48 h, relative luciferase activities were determined using the Dual-Luciferase Reporter Assay System (Promega, Madison, WI, USA) according to the manufacturer’s protocol. Relative activities were calculated by dividing the value of pRL by the value of pGL.

### 2.5. Pull-Down Analysis Using the Avidin–Biotin Binding System

BxPC-3 and HDF cells (4 × 10^5^ cells/1 mL/well) were seeded into 12-well microplates (4 × 10^5^ cells/1 mL/well) and preincubated overnight at 37 °C. The next day, cells were transfected with 5′-byotinylated single-strand RNA encoding miR-S1-3p or scrambled control (synthesized by Fasmac, Kanagawa, Japan) using the Dharmafect 1 reagent (GE Healthcare, Piscataway, NJ, USA). After 24 h, cells were lysed with avidin-pull-down lysis buffer (20 mM Tris-HCl pH 7.4, 100 mM KCl, 5 mM MgCl_2_, 0.5% NP-40, and 0.5 mM DTT) supplemented with RNaseOUT (Invitrogen) and pulled down with streptavidin-conjugated Dynabeads^TM^ magnetic beads (Invitrogen) for 1 h at 22–27 °C. RNAs from the resultant precipitate were isolated with ISOGEN (Nippon Gene, Osaka, Japan) and dis-solved with 15 µL of RNase-free H_2_O.

### 2.6. Immunoprecipitation (IP) by Anti-Ago1, Ago2, Ago3, and Ago4 Antibodies

The control and miR-S1-expressing BxPC-3 cells (7 × 10^5^ cells/2 mL/well) were seeded onto six-well microplates and preincubated overnight at 37 °C. The next day, cells were treated with 100 µM 4-thiouridine (Abcam, Cambridge, UK) for 16 h and exposed to 365 nm UV (0.15 J/cm^2^). After that, cross-linked cells were lysed with IP lysis buffer (25 mM Tris-HCl pH 7.4, 150 mM KCl, 0.5% NP-40, 20% glycerol, 1 mM NaF, 0.5 mM DTT, and 2 mM EDTA) supplemented with cOmplete ULTRA Tablets (Roche, Basel, Switzerland). After preclearance, these lysates were treated with 10 µg of monoclonal anti-Ago1 (clone 2A7, Wako, Osaka, Japan), Ago2 (clone 4G8, Wako, Osaka, Japan), Ago3 (clone 1C12, Wako, Osaka, Japan), and Ago4 (clone EPR23799-22, Abcam, Cambridge, UK) or isotype mouse IgG or rabbit IgG (Wako) for 2 h at 4 °C. Immunocomplexes were then precipitated with Protein G-conjugated Dynabeads^TM^ magnetic beads (Invitrogen) for 2 h at 4 °C, and RNAs from the resultant precipitate were isolated with ISOGEN (Nippon Gene, Osaka, Japan) and dis-solved with 15 µL of RNase-free H_2_O.

### 2.7. Quantitative Real-Time RT-PCR (RT-qPCR)

For the detection of pulled-down or immunoprecipitated RNA, cellular mRNA or miRNA, and SV40 levels in SV40-infected cells or in culture media, RT-qPCR was conducted. In all experiments, SYBR Premix Ex Taq II (Takara Bio Japan, Shiga, Japan) and the 7500 Fast Real-Time PCR System (Applied Biosystems, Foster City, CA, USA) were used. Detailed procedures of each RT-qPCR are described in the Appendix A and the primer sequences used in this procedure are listed in Appendix A.

### 2.8. Semiquantitative RT-PCR

Using an anchor RT primer, total RNAs (500 ng/reaction) were reverse transcribed with ReverTra Ace (TOYOBO, Osaka, Japan) following the manufacturer’s protocol. Using obtained cDNAs, semiquantitative RT-PCR by PrimeStar GXL (Takara Bio Japan) for LTag, STag, and tubulin (as an internal control) mRNA was performed. The thermocycling program was as follows: initial denaturation at 94 °C for 2 min; amplification at 98 °C for 10 s, 60 °C for 15 s, and 68 °C for 3 min; and final extension at 68 °C for 10 min of amplification. The RT-PCR products were separated by 0.6% agarose gel electrophoresis and visualized with a UV transilluminator. Using the Mighty TA cloning kit (Takara Bio), the sequences of the obtained PCR bands were confirmed. The primer sequences used in this experiment are listed in Appendix A.

### 2.9. Immunoblotting

Cells were lysed with 2× SDS sample buffer (50 mM Tris-HCl pH 6.8, 10% glycerol, 2% SDS, 0.1% bromophenol blue, and 0.2 M dithiothreitol) and the protein concentrations of the samples were quantified using a Bio-Rad Protein Assay Kit (Bio-Rad, Hercules, CA, USA). Subsequently, aliquots (12–15 µg of protein) were subjected to SDS-PAGE (8–12% gel) and transferred to PVDF membranes. An ECL prime Western blotting detection reagents (GE Healthcare) was applied to detect the signals. Detected signals were quantified using ImageJ software and their relative intensities were calculated. Information of the primary and secondary antibodies used in this experiment is described in the Appendix A. An ECL prime Western blotting detection reagents (GE Healthcare) was applied to detect the signals. Detected signals were quantified using ImageJ software and their relative intensities were calculated.

### 2.10. Telomeric Repeat Amplification Protocol (TRAP) Assay

Using cell pellets (2.5 × 10^5^), TERT activity was measured using the TRAPeze Telomerase Detection Kit (Merck Millipore, Darmstadt, Germany) according to the manufacturer’s protocol. Cells were lysed with CHAPS lysis buffer (10 mM Tris-HCl pH 7.5, 1 mM MgCl_2_, 1 mM EGTA, 0.1 mM benzamidine, 5 mM beta-mercaptoethanol, 0.5% CHAPS, and 10% glycerol) and incubated on ice for 30 min. After centrifugation (13,000× *g* at 4 °C for 20 min), aliquots of samples (equivalent to 50 cells in BxPC-3 and 80 cells in A549, respectively) were incubated at 30 °C for 30 min and telomeric repeats-added substrates (5′–AATCCGTCGAGCAGAGTT–3′) were amplified by PCR using Ex Taq polymerase (Takara Bio Japan). The PCR conditions were as follows: 95 °C for 2 min for initial denaturation, followed by 34 cycles of 94 °C for 15 s, 59 °C for 30 s, and 72 °C for 1 min for amplification. PCR products were then subjected to nondenatured PAGE (12% gel), stained with 1 µg/mL ethidium bromide (Promega), and gel images were obtained using an ImageQuant LAS4000 (GE Healthcare). Detected signals were quantified using ImageJ software and their relative intensities were calculated.

### 2.11. Viral Infection Experiments

SV40 and miR-S1-deficient SV40 (SV40/TAD) were used for further experiments. All experiments used in them were performed in a research institution of biosafety level 2 (BSL-2), Musashino University. Before bringing out all samples from the BSL-2 area, heat inactivation (95 °C for 10 min) was carried out.

### 2.12. SV40 and SV40/TAD Preparation

To prepare the SV40 or SV40/TAD, plasmids containing the SV40 or SV40/TAD genome (pMK16-wSV40 or pMK16-SV40/TAD) [16] were digested with *Bam* HI and each genome region was self-ligated. The resulting circular SV40 or SV40/TAD was transfected into CV-1 cells for seven days, then passed to COS-7 cells, and incubated for another two weeks. Harvested supernatants were centrifuged at 10,000× *g* for 5 min to remove cellular debris, and their titers were examined using PCR.

Prepared SV40 and SV40/TAD were used for infection to control or miR-1266-expressing A549 cells. The infection procedure was as follows: Cells were seeded onto 12-well microplates (5 × 10^4^ cells/well) and preincubated overnight at 37 °C. Subsequently, cells were infected with wild-type SV40- or SV40/TAD-contained CV-1 cultured medium at 10^5^ particles. After 2 h of viral exposure, the culture medium was replaced with a fresh one and incubated for 24 or 72 h. These obtained cells and culture media were subjected to RT-qPCR, luciferase reporter assays, immunoblotting, and TRAP assays as described above.

### 2.13. Statistical Analysis

All comparisons were performed using two-tailed Student’s *t*-test, and differences were considered statistically significant at a *p* value of < 0.05.

## 3. Results

### 3.1. MiR-S1-3p Directly Binds to Cellular miR-1266-5p

To identify human miRNAs potentially bound to viral miR-S1-3p by base-paring complementarity, we employed the miRBase database (http://www.mirbase.org/, accessed on 10 July 2022) to search for miRNAs with sequence similarity to miR-S1-5p, which shares considerable complementarity with miR-S1-3p in the miR-S1 duplex. Consequently, we identified nine putative miRNAs interacting with miR-S1-3p, such as miR-138-5p, miR-152-3p, miR-337-5p, miR-434-5p, miR-532-3p, miR-921, miR-1266-5p, miR-4261, and miR-6771-5p (Figure 1A). Next, these miRNA candidates were assessed for their hybridization with miR-S1-3p by predicting the secondary structure of the interacting duplexes and their minimal free energy (Figure 2). The model secondary structure proposed for the candidate miRNA sequence-GAAA-miR-S1-3p sequence or miR-S1-3p sequence-GAAA-the candidate sequence was generated using a folding algorithm, namely, mfold software (http://www.unafold.org, accessed on 10 July 2022). RNAstructure software (https://rna.urmc.rochester.edu/RNAstructure.html, accessed on 10 July 2022) was used to analyze data from the candidate miRNA sequence-GAAA-miR-S1-3p sequence or miR-S1-3p sequence-GAAA-the candidate sequence. The minimal free energy values predicted for the hybridization of miR-1266-5p and miR-6771-5p with miR-S1-3p are −25.5 kcal/mol/−24.8 kcal/mol and −27.2 kcal/mol/−26.2 kcal/mol, respectively, which are much less than those of the canonical miR-S1-3p–miR-S1-5p duplex (–25.5 kcal/mol/–24.8 kcal/mol). These analyses suggest that miR-1266-5p or miR-6771-5p forms a stable RNA duplex with miR-S1-3p.

To evaluate whether miR-S1-3p can bind to the nine miRNA candidates, we employed a luciferase reporter assay using the reporter plasmids containing an miRNA candidate sequence in the 3′ UTR of the *luciferase* gene. Upon the transfection of pre-miR-S1, the reporter activity was significantly reduced in the constructs containing miR-138-5p, miR-152-3p, miR-337-5p, miR-1266-5p, and miR-6771-5p sequences (Figure 1B), suggesting that these five miRNAs interact with miR-S1, and possibly miR-S1-3p. On the basis of the present results of the stability of the miRNA/miRNA complex as well as the reporter experiments, we focused on miR-1266-5p in the following study.

A biotin–avidin pull-down test in BxPC-3 cells and human dermal fibroblasts (HDF) revealed the direct association of miR-S1-3p with endogenous miR-1266-5p. These cells were transfected with biotinylated miR-S1-3p and their cell lysates were precipitated via avidin-conjugated magnetic beads, followed by the RT-qPCR analysis of the precipitated complex. A significant amount of endogenous miR-1266-5p was coprecipitated by biotinylated miR-S1-3p as compared with biotinylated control miRNA (scramble) in both cells. (Figure 3A). However, mutated miR-S1-3p with five mismatches in the 3′ end or six mismatches in the 5′ end reduced the amount of coprecipitated miR-1266-5p to control levels observed in the scramble miRNA (Figure 3A).

Although miR-S1 intracellularly interacts with miR-1266-5p, overexpression of miR-S1 into BxPC-3 cells did not lead to the downregulation of endogenous miR-1266-5p (Figure 3B). This prompts us to examine whether the interaction of miR-S1 with miR-1266 occurs within the RNA-induced silencing complex (RISC). An anti-Ago2 antibody was used with immunoprecipitated lysates from cells transfected with miR-S1 expression or control plasmids. On the RNA from these immunoprecipitated complexes, RT-qPCR was performed to determine the levels of miR-1266-5p and miR-S1. Significant enrichment of miR-1266-5p and miR-S1 was detected in the complexes precipitated with the Ago2 antibody (Figure 3B), suggesting their interactions within the RISC containing Ago2.

### 3.2. miR-S1-3p Prevents miR-1266-5p from Downregulating TERT

TERT has been experimentally shown to be a target of miR-1266 [17]. This was confirmed by reporter assays and immunoblotting. The luciferase reporter plasmids harboring the 3′ UTR of *hTERT* (Luc-hTERT-3′ UTR) were cotransfected with miR-1266 expression or control plasmids into HEK293 cells. Forced miR-1266 expression significantly reduced the luciferase activity of Luc-hTERT-3′ UTR compared with the control. However, the mutation of an miR-1266 binding site within *hTERT* 3′ UTR (Luc-hTERT-3′ UTRmt) restored the luciferase activity almost to the level of the control (Figure 4A). For immunoblotting analyses, plasmids expressing the HA-tagged hTERT gene, including its 3′ UTR and either miR-1266 or control plasmids, were transfected into HEK293 cells. The lysate of the miR-1266-transfected cells contained less HA-tagged hTERT protein than that of control-transfected cells.

To examine whether miR-S1 can counteract the miR-1266-induced downregulation of TERT by its sponge activity for miR-1266, we evaluated the miR-S1 effects on TERT expression by reporter assays. The reporter plasmids of Luc-hTERT-3′ UTR were cotransfected with miR-1266 or miR-S1 or both into HEK293 cells, followed by the measurement of luciferase activity. Although miR-S1 expression alone did not alter luciferase activity, its coexpression with miR-1266 resulted in counteracting the inhibitory effects of miR-1266 on TERT luciferase activity (Figure 4B). This counteraction of miR-S1 against miR-1266 was evaluated by RT-qPCR analyses of TERT transcripts within AGOs-containing immunoprecipitants. The lysates of BxPC-3 cells transfected with miR-S1 or control plasmids were immunoprecipitated by anti-Ago1-4 antibodies, core components of the RISC. The subsequent quantification of *TERT* mRNAs in the Ago1-4-containing immunocomplexes by RT-qPCR revealed that *TERT* mRNAs in Ago2-containing RISCs were decreased by miR-S1 expression compared with control, whereas their amounts in Ago1, 3, and 4-containing RISCs did not represent any differences between miR-S1 and control (Figure 4C). These results suggest that miR-S1 expression reduces the recruitment of an miR-1266 target or *TERT* mRNA into the Ago2-containing RISCs.

Furthermore, we validated the miR-S1 effects on the level of endogenous TERT proteins expressed in BxPC-3 and A549 cells. MiR-S1 overexpression resulted in a slight increase in TERT proteins in BxPC-3 cells but not in A549 cells (Figure 4D). In these cells, however, miR-1266 overexpression induced a decrease in TERT proteins, which was recovered by miR-S1 expression. Consistently, similar results were also observed in the analysis of TERT activity using the TRAP assay. TERT activity in BxPC-3 cells was slightly increased, and it recovered the TERT activity reduced by miR-1266 expression (Figure 4E).

### 3.3. SV40 Infection Induced Mutual Sequestration between Viral miR-S1 and Cellular miR-1266 to Derepress Their Respective Targets, Telomerase and T Antigens

Forced miR-S1-3p expression by transfection was found to prevent miR-1266-5p from repressing telomerase, implying the possibility that miR-1266-5p conversely sponges miR-S1 activity just to derepress its targets, or T antigens. To evaluate this possibility, SV40 was infected with control and miR-1266-expressing A549 cells, in which the expression levels of telomerase as well as T antigens were examined by dual-luciferase reporter assay, RT-qPCR, and immunoblotting analyses. Prior to these examinations, the expression of miR-S1 and miR-1266 in the infected cells was verified by RT-qPCR. As expected, wild-type SV40 (wtSV40)-infected miR-1266-expressing A549 cells represented the abundant expression of both miR-S1 and miR-1266, which could interact with each other, and wtSV40-infected control cells and uninfected miR-1266-expressing A549 cells represented the sole expression of miR-S1 and miR-1266, respectively (Figure 5A).

Dual-luciferase reporter assays revealed that wtSV40-infected miR-1266-expressing cells exhibited a slightly enhanced activity of a reporter harboring a target site for miR-S1 in the *T antigen* mRNA (Luc-Tag-miR-S1) compared to wtSV40-infected control cells (Figure 5B). These infected control cells displayed less activity than uninfected cells. However, mutating the miR-S1 target site in the reporter abolished not only the suppressive effects of wtSV40 infection on the reporter activity in control cells but also its enhancing effects in miR-1266-expressing cells, suggesting that these effects involve miR-S1 induced by wtSV40 infection. Similarly, the activity of a reporter containing *TERT* 3′ UTR (Luc-hTERT-3′ UTR) was enhanced by wtSV40 infection in miR-1266-expressing cells when compared to its infected control cells (Figure 5B). However, uninfected miR-1266-expressing cells exhibited less reporter activity than uninfected control cells. Moreover, the mutation of an miR-1266 target site in the reporter abolished the enhancing and repressive effects. Taken together, the present results indicated that the coexpression of miR-S1 and miR-1266 in infected cells leads to the increasing activity of their target reporters, namely, Tag and telomerase, although miR-S1 or miR-1266 alone suppresses their respective reporter activity.

Consistently, the RT-qPCR analyses also revealed that wtSV40-infected miR-1266-expressing cells contained more transcripts of large and small T antigen transcripts (*LSTag*) and *TERT* than wtSV40-infected control cells did (Figure 5C,D). However, SV40/TAD (a mutated strain lacking miR-S1 expression) infection conferred a similar level of *LSTag* and *TERT* on control and miR-1266-expressing cells, suggesting that miR-S1 expression induces an increase in *LSTag* and *TERT* in miR-1266-expressing cells. Because the RT-qPCR analyses were unable to evaluate individual transcripts of large and small T antigen (*LTag* and *STag*), these two transcripts were individually assessed by semiquantitative PCR analyses using distinct primer sets with the verification of PCR products by sanger sequences (Figure 5C). These PCR analyses also showed that *LTag* and *STag* were more abundant in wtSV40-infected miR-1266-expressing cells than in the infected control cells. Moreover, immunoblotting analyses demonstrated that wtSV40 infection induced significantly increased protein expression of small T antigen in miR-1266-expressing cells, but had only marginal effects on the expression of large T antigen and TERT (Figure 5E). Taking intracellular miR-S1–miR-1266 interactions into account, the present results of RT-qPCR and immunoblotting analyses revealed that miR-S1 and miR-1266 sequester each other from their respective targets to derepress the expression of T antigens and TERT.

### 3.4. SV40 Infection-Induced miR-S1–miR-1266 Interplay Enhanced Telomerase Activity and Viral Replication

Infection-induced mutual sequestration between miR-S1 and miR-1266 led to the derepressed expression of TERT and T antigens, which were involved in cellular transformation and viral replication. These findings prompted us to assess TERT activity and viral replication in wtSV40-infected miR-1266-expressing A549 cells, which were compared to uninfected miR-1266-expressing cells in TERT activity or wtSV40-infected control cells in viral replication.

TERT activity was quantified by the densitometric analyses of the PCR products obtained from the TRAP assay (Figure 6A). These quantifications revealed that wtSV40-infected miR-1266-expressing cells exhibited higher TERT activity than uninfected miR-1266-expressing cells, whose TERT expression was downregulated by miR-1266, as shown in the present experiment (Figure 4E). However, wtSV40 infection conferred a similar TERT activity on control and miR-1266-expressing cells (Figure 6A). These data show that miR-S1 acts as a sponge for miR-1266′s repressive effects on TERT activity. qPCR studies of SV40 genomic DNA included in infected cells and their culture media, which were extracted at one day postinfection (dpi) and three days postinfection (dpi), were used to assess viral replication. The fold changes of viral genome DNA were calculated by normalizing the amounts of viral DNA at 3 dpi to those at 1 dpi. These analyses indicated that miR-1266-expressing A549 cells more efficiently produce intracellular and extracellular SV40 DNA than control A549 cells, suggesting that miR-S1–miR-1266 interplay promotes viral replication and release. In both control and miR-1266-expressing A549 cells, however, SV40/TAD infection employed lower viral replication and release compared to wtSV40 infection (Figure 6B). These results were consistent with previous studies [16,18]. The impaired replication of SV40/TAD lacking miR-S1 expression may be attributed to the sequestration of cellular proteins required for viral DNA replication from the proper replication sites by excessive large T antigens, which interact with the cellular proteins.

## 4. Discussion

The present study has expanded our understanding of and provided mechanistic insight into the functional roles of SV40 miR-S1 in the context of SV40-induced cell transformation and viral replication. Viral miR-S1 and cellular miR-1266 interact with each other just to sequester each other from their targets, resulting in the increased expression of not only viral small and large T antigens but also a cellular protein, TERT. Because small T antigen activates TERT, this simultaneous increase in production of small T antigen and TERT proteins would promote TERT activity synergistically (Figure 6A) [19,20]. TERT activity may also be enhanced by the sequestration of miR-138-5p, which interacts with miR-S1 and negatively regulates TERT expression (Figure 1B) [21]. Conversely, the elevated expression of large T antigens contributed to the promotion of SV40 replication in miR-1266-expressing A549 cells. Collectively, the present study highlights a novel interplay between SV40-miR-S1 and cellular miRNAs, which underlies miR-S1 functions in cell transformation and viral replication.

Viral miRNAs, in addition to viral proteins, have a role in cell transformation and oncogenesis. Various viral miRNAs bind to host mRNAs and suppress the expression of biological proteins that regulate cell proliferation and/or survival, facilitating cell transformation and oncogenesis [22,23,24,25,26,27]. Meanwhile, some miRNAs encoded in polyomaviruses and herpesviruses downregulate viral oncoproteins, such as T antigens and LMP1 [8,9,28]. These miRNAs appear to attenuate the ability of the viral oncoproteins to promote transformation and oncogenesis. However, this is not the case when infected cells abundantly express tumor suppressor miRNAs, which target not only cellular oncoproteins but also the viral miRNAs. In these types of cells, both viral and cellular oncoproteins could be derepressed by the mutual sequestration between the viral and cellular miRNAs, possibly leading to the promotion of oncogenesis. In this context, it is intriguing to know that miR-1266, found as an miR-S1-interacting miRNA, suppresses cell growth, metastasis, and invasion in prostate, gastric, and papillary thyroid cancers [17,29], and it promotes cell proliferation, migration, and invasion in cervical cancers and hepatocellular carcinoma, etiologically associated with human papillomavirus and hepatitis B virus infection [30,31]. In virus-induced cervical malignancies and hepatocellular carcinoma, it may be worthwhile to investigate the physical and functional interactions between viral miRNAs and cellular tumor suppressor miRNAs.

In addition to miR-1266, miR-138-5p and miR-152-3p are found to interact with miR-S1, and therefore, the interactions may contribute to cellular transformation and oncogenesis because they have been reported as tumor suppressor miRNAs in various types of cancer [32,33,34,35]. In addition to *TERT*, miR-1266 targets *BCL2* to induce apoptosis, and it targets *FGFR2* to attenuate cell proliferation, invasion, and migration [27,36]. miR-138-5p targets *FOXC1* and *DEK* to inhibit the progression of prostate cancer and proliferation of gastric cancer, respectively [33,37,38]. Surprisingly, some long noncoding RNAs sponge miR-152-3p, ultimately promoting tumorigenesis in multiple myeloma and colorectal cancers [39,40,41,42]. A similar sponging phenomenon would be possible between miR-S1 and miR-152-3p to promote tumorigenesis. Therefore, as to the miR-S1-interacting miRNAs, we are going further to explore the functional roles of miR-S1, via an interplay between it and cellular miRNA, in cell transformation and tumorigenesis.

Viral and cellular microRNAs may be co-operatively involved in the control of viral latent–lytic switches. Polyomavirus and herpesvirus microRNAs negatively regulate the expression of viral transactivating proteins involved in replications [3,9,43,44,45,46,47]. They demonstrate that their downregulation of the viral transactivating proteins leads to the establishment of viral latency, therefore proposing a role for viral microRNAs in the latent–lytic switch [48,49,50]. Contrarily, the sequestration of such viral microRNAs by cellular microRNAs may result in the shifting of viral latent–lytic switches to the induction of lytic replication. Briefly, the availability of cellular microRNAs that sequester the viral microRNAs in infected cells determines whether the infection is latent or lytic. Indeed, all herpesviruses can produce lytic or latent infections depending on the host cell type [51]. These viral infections in a cell type-specific manner are attributed to not only cellular and viral proteins directly or indirectly involved in transcription but also cellular and viral microRNAs [52].

In conclusion, we found a novel mechanism underlying the functional roles of miR-S1 to modulate not only cellular transformation and oncogenesis but also viral replication. The present finding of a novel interplay between viral and cellular microRNAs will further point to and will elucidate the important roles of polyomavirus and herpesvirus miRNAs in virus-induced pathogenesis. Furthermore, it provides a novel notion that viral lytic or latent infection in a cell type-specific manner is attributable to cellular miRNAs as well as transcriptional factors.

## Figures and Tables

**Figure 1 ncrna-08-00057-f001:**
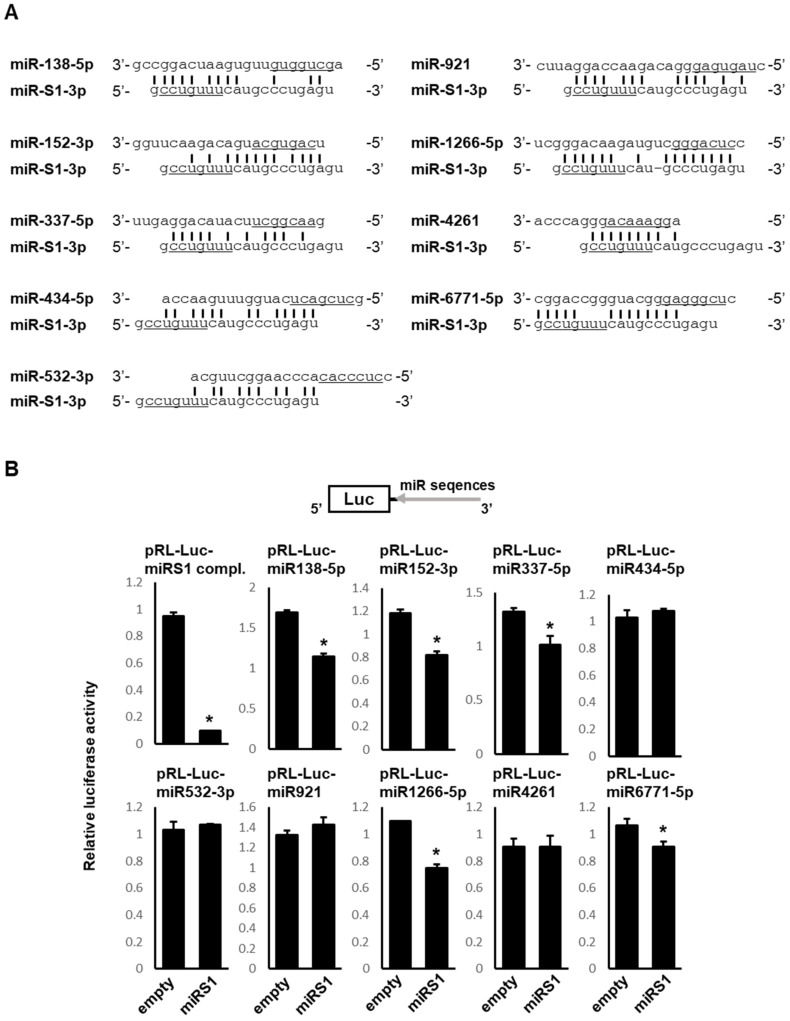
MiR-138-5p, miR-152-3p, miR-337-5p, miR-1266-5p, and miR-6771-5p are targets for SV40-miR-S1. (**A**) Schematic representation of possible complementary forms between miR-S1-3p and miR-138-5p, miR-152-3p, miR-337-5p, miR-434-5p, miR-532-3p, miR-921, miR-1266-5p, miR-4261, and miR-6771-5p. Seed sequences of each miRNA are underlined. (**B**) HEK293 cells were cotransfected with an internal pGL control vector, reporter vectors (pRL-Luc-miR-S1 compl., pRL-Luc-miR-138-5p, pRL-Luc-miR-152-3p, pRL-Luc-miR-337-5p, pRL-Luc-miR-434-5p, pRL-Luc-miR-532-3p, pRL-Luc-miR-921, pRL-Luc-miR-1266-5p, pRL-Luc-miR-4261, and pRL-Luc-miR-6771-5p), and mammalian expression vectors encoding pre-miR-S1 (CMV-miR-S1-PGK-puro) at a molar ratio of 1:5:25. pRL-Luc-miR-S1 compl. (described in [16]) was used as a positive control. The relative luciferase activities were calculated by dividing the value of pRL by the value of pGL. All values were normalized by empty vector-transfected cells. All experiments were repeated three times. Columns, mean values (*n* = 3); bars, standard deviation (SD); * *p* < 0.05.

**Figure 2 ncrna-08-00057-f002:**
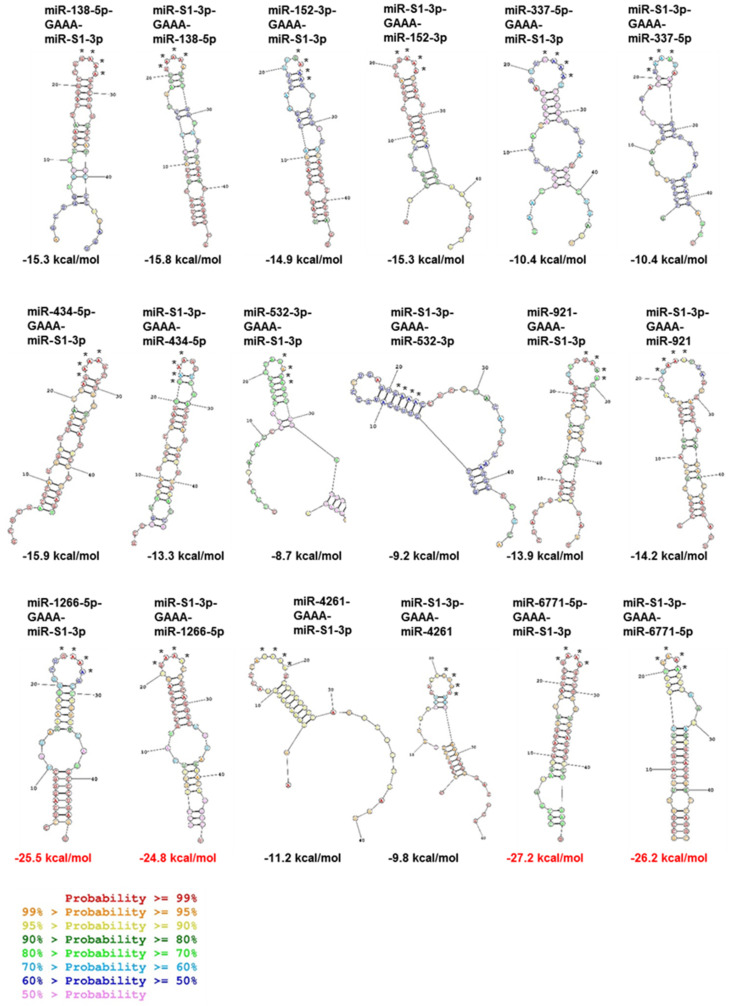
Secondary structures of duplexes between miR-S1-3p and miR-138-5p, miR-152-3p, miR-337-5p, miR-434-5p, miR-532-3p, miR-921, miR-1266-5p, miR-4261, and miR-6771-5p. Each duplex was predicted using RNAstructure software (https://rna.urmc.rochester.edu/RNAstructure.html, accessed on 10 July 2022). Before outputting secondary structures, each combination of miRNAs was aligned by the GAAA linker sequence. The Gibbs free energies of predicted duplexes were shown under each secondary structure. Colors of each nucleotide symbol shows the probability that it will form the miR-S1-subjected miRNA-duplex. * Position of GAAA linker sequences.

**Figure 3 ncrna-08-00057-f003:**
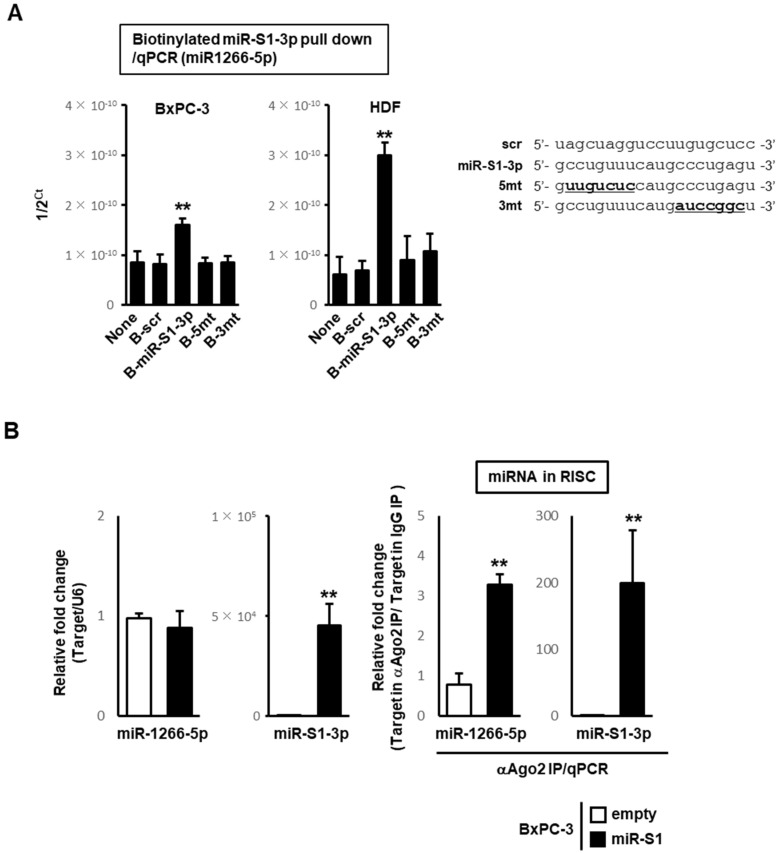
MiR-S1-3p directly binds to endogenous miR-1266-5p, and trapped miR-1266-5p is sequestered in Ago2-associated RISC. (**A**) Lysates from biotinylated scramble sequence, miR-S1-3p, 5mt, and 3mt-transfected cells were pulled down with streptavidin-conjugated magnetic beads, and then RNAs were isolated. Using these RNAs, the levels of miR-1266-5p were determined by RT-qPCR. The Sequence of miR-S1-3p, 5mt, and 3mt (underlined) was also shown (right). All experiments were repeated two times. Columns, mean values (*n* = 3); bars, SD; ** *p* < 0.01. (**B**) Total RNAs from control and miR-S1-expressing BxPC-3 were subjected to RT-qPCR for miR-1266-5p and miR-S1-3p, respectively. In parallel, lysates from control and miR-S1-expressing BxPC-3 were treated with anti-Ago2 antibodies, precipitated with protein G-conjugated magnetic beads, and then RNAs were isolated. Using these RNAs, their levels of miR-1266-5p and miR-S1-3p were determined by RT-qPCR. All experiments were repeated two times. Columns, mean values (*n* = 3); bars, SD; ** *p* < 0.01.

**Figure 4 ncrna-08-00057-f004:**
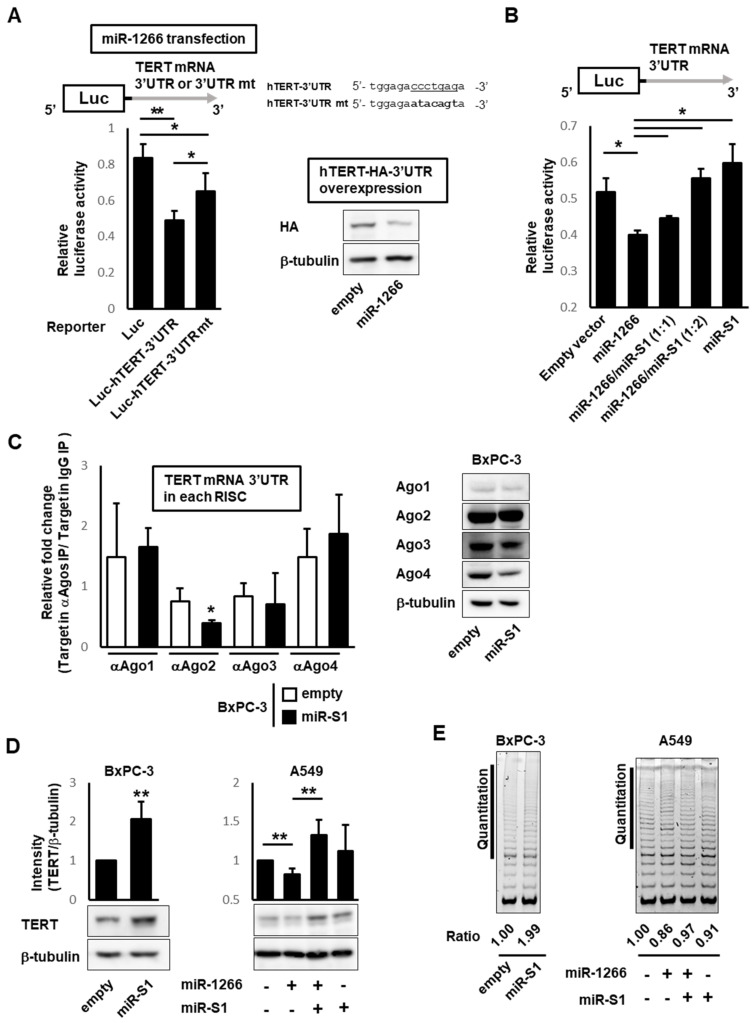
MiR-S1 upregulates TERT levels through competitive antagonism with miR-1266. (**A**) HEK293 cells were cotransfected with the internal pGL control vector and pRL-Luc, pRL-Luc-hTERT-3′ UTR, or pRL-Luc-hTERT-3′ UTR mt, and CMV-miR-1266-PGK-puro at a molar ratio of 1:5:25. The relative luciferase activities were calculated by dividing the value of pRL by the value of pGL. All values were normalized by empty CMV-cont-PGK-puro-transfected cells. The sequence of the mutated region of pRL-Luc-hTERT-3′ UTR mt was also shown (underlined). Further, control and miR-1266−expressing cells were further transfected with pZac-hTERT-HA-3′ UTR. The levels of HA in the resultant transfectants were determined by immunoblot analysis. Each experiment was repeated two (immunoblot analysis) or three (reporter assay) times. Columns, mean values (*n* = 3); bars, SD; * *p* < 0.05; ** *p* < 0.01. (**B**) HEK293 cells were cotransfected with the internal pGL control vector, pRL-Luc-hTERT-3′ UTR, and miRNA expression vectors at a total molar ratio of 1:5:25. The relative luciferase activities were calculated by dividing the value of pRL by the value of pGL. All values were normalized by corresponding empty vector-transfected cells. All experiments were repeated three times. Columns, mean values (*n* = 3); bars, SD; * *p* < 0.01. (**C**) Lysates from control and miR-S1-expressing BxPC-3 were also treated with anti−Ago1, Ago2, Ago3, and Ago4 antibodies before being precipitated with protein G-conjugated magnetic beads, and RNAs were isolated. Using these RNAs, the level of TERT mRNA 3′ UTR was determined by RT-qPCR. Columns, mean values (*n* = 3); bars, SD; * *p* < 0.05. The levels of Ago1, Ago2, Ago3, and Ago4 in control and miR-S1-expressing BxPC-3 were also determined by immunoblot analysis. (**D**) Levels of endogenous TERT protein in BxPC-3- and A549-derived infectants were determined using immunoblot analysis. The levels of TERT protein (TERT/beta-tubulin) were calculated using ImageJ software. All experiments were repeated three times. Columns, mean values (*n* = 4); bars, SD; ** *p* < 0.01. (**E**) Cellular TERT activities in BxPC-3− and A549−derived infectants were measured using the TRAP assay. TRAP reactions were conducted by 50 cells (BxPC-3) or 80 cells (A549) per reaction, respectively. Using ImageJ software, specific activities were calculated by quantifying the intensity of ladder bands at vertical lines (middle−long telomeric repeats). All experiments were repeated two times.

**Figure 5 ncrna-08-00057-f005:**
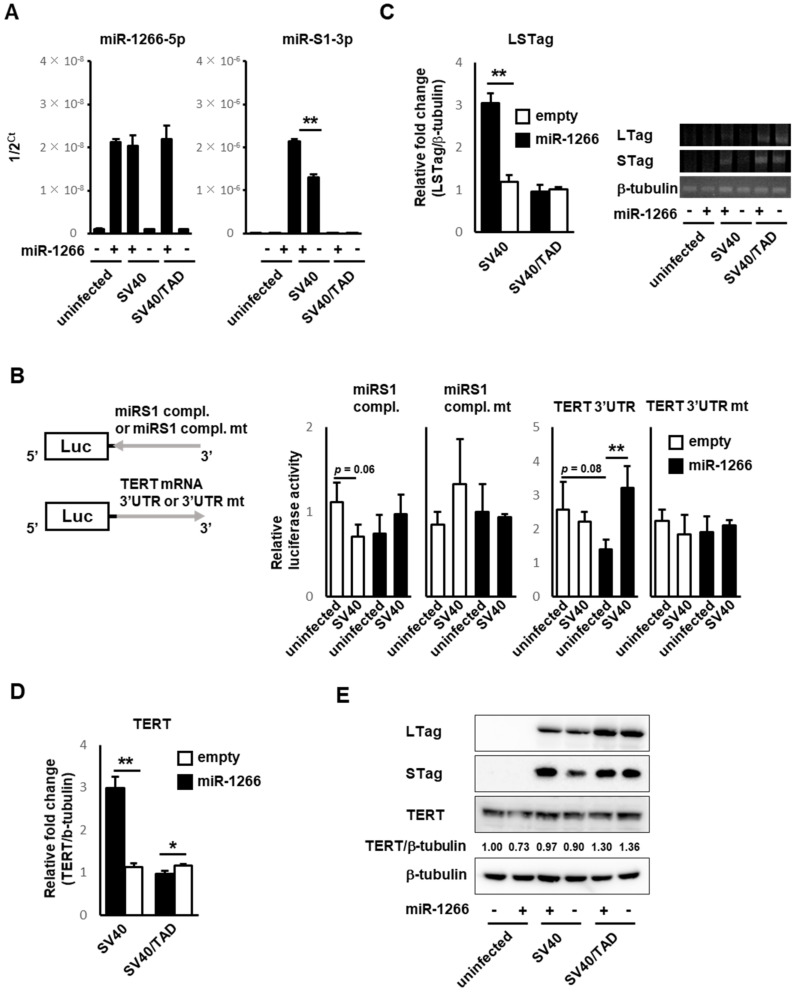
MiR-S1 induced by SV40 infection and miR-1266 mutually interfere with their respective targets, namely, TERT and T antigens. (**A**,**C**) Total RNAs were extracted from control and miR-1266 −expressing A549 cells infected with SV40 or SV40/TAD. At 3 dpi, total RNAs were subjected to RT-qPCR for miR-1266-5p and miR-S1-3p. Levels of miR-1266-5p, miR-S1-3p, and LSTag were determined by RT-qPCR. Levels of LTag and STag mRNA were also sorely detected by semiquantitative RT-PCR. All experiments were repeated two times. Columns, mean values (*n* = 3); bars, SD; ** *p* < 0.01. (**B**) Uninfected or SV40-infected control and miR-1266−expressing A549 cells were cotransfected with internal pGL control vector and pRL-Luc, pRL-Luc-miR-S1 compl., pRL-Luc-miR-S1 compl. mt, pRL-Luc-hTERT-3′ UTR, or pRL-Luc-hTERT-3′ UTR mt at a molar ratio of 1:5. The relative luciferase activities were calculated by dividing the value of pRL-Luc by the value of pGL. All experiments were repeated two times. Columns, mean values (*n* = 3); bars, SD; ** *p* < 0.01. (**D**) Total RNAs from control and miR-1266−expressing A549 were infected with SV40 or SV40/TAD. At 3 dpi, total RNAs were subjected to RT-qPCR for TERT mRNA. Columns, mean values (*n* = 3); bars, SD; * *p* < 0.05; ** *p* < 0.01. (**E**) Levels of LTag, STag, and TERT proteins in uninfected or SV40-infected control and miR-1266−expressing A549 cells were determined using immunoblot analysis. Expression ratios of TERT protein (TERT/beta-tubulin) were calculated using ImageJ software. All experiments were repeated two times.

**Figure 6 ncrna-08-00057-f006:**
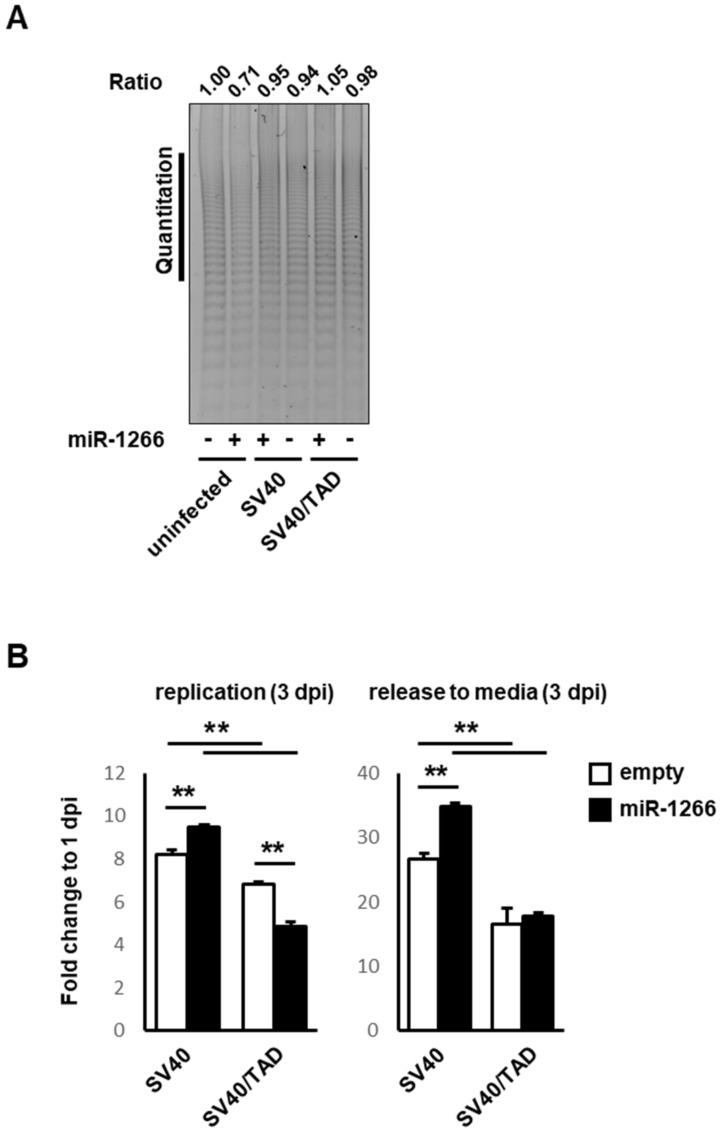
The interplay between miR-S1 and miR-1266 promotes TERT activity and viral replication of SV40. (**A**) Using the TRAP assay, cellular TERT activities in uninfected, SV40, or SV40/TAD-infected control and miR-1266−expressing A549 cells were determined. TRAP reactions were conducted by 80 cells per reaction. Using ImageJ software, specific activities were calculated by quantifying the intensity of ladder bands at vertical lines (middle−long telomeric repeats). All experiments were repeated two times. (**B**) Control and miR-S1−expressing A549 were infected with SV40 or SV40/TAD, and then genomic DNA from cells or culture media was subjected to RT-qPCR with the specific primers for LSTag. Fold changes of the SV40 level at 3 dpi were determined by dividing them by 1 at 1 dpi. All experiments were repeated two times. Columns, mean values (*n* = 3); bars, SD; ** *p* < 0.01.

## Data Availability

Not applicable.

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
