# Peer review of "SV40 miR-S1 and Cellular miR-1266 Sequester Each Other from Their Targets, Enhancing Telomerase Activity and Viral Replication"

_ncrna, 2022, doi:10.3390/ncrna8040057_

Round 1

Reviewer 1 Report

The manuscript was written in a logical manner and shown a novel role of the viral derived miR-S1 as a sponge to sequester miR-1266 for viral replication. However, the authors need to revise the manuscript extensively because there are many spelling errors throughout the whole manuscript. 

1) Why do the authors decide to overexpress miR-1266 in A549 cells as a model to study the interactions of miR-S1 with miR-1266 instead of using a cell line that naturally expresses miR-1266 (i.e. BxPC-3 or HDF)?

2) The authors should explain more on why overexpression of miR-1266 suppresses TERT 3'UTR  in figure 5B. Does miR-1266 induce apoptosis in A549 cells since it is an tumor suppressor miRNA?

Author Response

  • Why do the authors decide to overexpress miR-1266 in A549 cells as a model to study the interactions of miR-S1 with miR-1266 instead of using a cell line that naturally expresses miR-1266 (i.e. BxPC-3 or HDF)?

Before performing viral infection experiments using A549, the preliminary test whether A549, BxPC-3, and HDF can be infected by SV40 was conducted. As a result, adequate efficacies of SV40 infection in BxPC-3 and HDF were not observed and we judged that A549 is most suitable line for our viral infection experiments.

  • The authors should explain more on why overexpression of miR-1266 suppresses TERT 3'UTR in figure 5B. Does miR-1266 induce apoptosis in A549 cells since it is an tumor suppressor miRNA?

We are worrying that you recognize Fig. 5B as the results of RT-qPCR. The results of Fig. 5B shows reporter activities of Luciferase contained TERT 3’ UTR at downstream of 3’ end of Luciferase gene. In Fig. 4, effect of miR-1266 on interactions to TERT 3’ UTR have been described in detail.

During our experiments in this study, a number of opportunities that miR-1266-expressing A549 was observed. At least, obvious apoptotic induction by miR-1266 expression was not found microscopically, so we have not assessed apoptosis assays in this study.

Finally, we thank you for important indication about spelling misses of our manuscript. We checked carefully our manuscript once again and corrected them.

Reviewer 2 Report

The article entitled “SV40 miR-S1 and cellular miR-1266 sequester each other from their targets, enhancing telomerase activity and viral replication” by Takahashi et al points out a novel mechanism of a SV40 viral microRNA (miR) miR-S1 interaction that regulates host cellular machinery for its effective viral replication. A very few studies have shown miR-miR interaction, followed by the downstream mRNA targets and protein synthesis dynamics effected by such an event. Here, the authors describe the interaction of miR-S1 to its host microRNA miR-1266, in a series of well-designed workflow. At first, they find the miRNA binding affinity (complementarity) between viral and host miRNAs, they then perform luciferase reporter assay to confirm the miR-S1 interaction with candidate miRNAs and adopted miR-1266 as a potential miRNA target (among four other significant miRNAs) for the rest of the analysis. Upon confirming the strong association between miR-S1 and miR-1266 in Ago2-RISC complex, the targets of miR-1266 mainly TERT was examined. Overall, the manuscript is well drafted and I feel that the manuscript is suitable for publication after the following issues are addressed:

Minor

  1. microRNAs are known to target more than one mRNAs; however, a handful of experiments demonstrate miRNAs targeting miRNAs. The overall analysis proves that the miR-S1 targets miR-1266-5p, which is a mature and dominant miRNA compared to its 3p counterpart. What is the site of SV40-infection, i.e., across which cell type? The expression of miR-1266 which has been characterized and proven to be the potential target in this research has substantially low levels of expression in various primary and cancer cell types (<50 CPM)1, tissues and organs (< 10 RPM)2. This is one of the concerns that I have considering future research in this direction, i.e., Is miR-1266 any contributor to the mechanism in miR-S1 induced virulence in-vivo? Addressing this issue considering the levels of miR-1266 expression in various primary and cancer cells as a note in the main manuscript is suggested. 

1 De Rie et al., (2017). An integrated expression atlas of miRNAs and their promoters in human and mouse. Nature Biotechnology. 35, 872–878.

2 Keller et al., (2022). miRNATissueAtlas2: an update to the human miRNA tissue atlas. Nucleic Acids Research, 50, D1, D211–D221. 

  1.  Line 21: “induced by miR-These effects of” and line: 255 “and miR-SSignificant enrichment”, are potential technical errors and need to be corrected throughout the manuscript for similar errors.
  2. The Figures need to be appropriately numbered in the Figure legends. Except for Figure 1, no other figures have numbers.
  3. Citation of the use of resources such as databases (miRBase) and software tools (Unfold, RNAstructure) are missing.

Author Response

Reply to reviewer 2

  1. microRNAs are known to target more than one mRNAs; however, a handful of experiments demonstrate miRNAs targeting miRNAs. The overall analysis proves that the miR-S1 targets miR-1266-5p, which is a mature and dominant miRNA compared to its 3p counterpart. What is the site of SV40-infection, i.e., across which cell type? The expression of miR-1266 which has been characterized and proven to be the potential target in this research has substantially low levels of expression in various primary and cancer cell types (<50 CPM)1, tissues and organs (< 10 RPM)2. This is one of the concerns that I have considering future research in this direction, i.e., Is miR-1266 any contributor to the mechanism in miR-S1 induced virulence in-vivo? Addressing this issue considering the levels of miR-1266 expression in various primary and cancer cells as a note in the main manuscript is suggested. 

SV40 preferentially infects kidney, liver, and lung and so on. As you pointed out, these and other tissues have low level of miR-1266 expression level. This is one of the reason why we used miR-1266-overexpressing cells in viral infection experiments. However, several reports showed the clinical influence on up- or downregulation of miR-1266 in cancer and skin disease (See ref. 16, Ichihara et al., Eur. J. Dermatol., 22, 68-71, 2012, and Huang et al., Oncol. Lett., 21, 347, 2022). Therefore, we still believe that miR-1266, together with other miR-S1-interacting miRNAs, can be an effector of miR-S1-induced virulence in some diseases described above, although it may have only marginal effects. Nevertheless, we should fully consider the expression level of miR-S1-interacting miRNAs with the cells, when the future study of pathogenesis caused by miR-S1/miRNAs interaction will be performed. 

 Line 21: “induced by miR-These effects of” and line: 255 “and miR-SSignificant enrichment”, are potential technical errors and need to be corrected throughout the manuscript for similar errors.

Thank you for important indication. We checked carefully our manuscript once again and corrected spelling misses including example you demonstrated.

2. The Figures need to be appropriately numbered in the Figure legends. Except for Figure 1, no other figures have numbers.

Thank you for important indication. We checked spelling misses and added appropriate numbers in Figure legends.

3. Citation of the use of resources such as databases (miRBase) and software tools (Unfold, RNAstructure) are missing.

Thank you for important indication. We checked and corrected links for databases in the manuscript.